


# 1  Peatlands and their carbon dynamics in northern high latitudes from
# 2  1990 to 2300: A process-based biogeochemistry model analysis

Bailu Zhao[1], Qianlai Zhuang[1, 2]
[1] Department of Earth, Atmospheric, and Planetary Sciences, Purdue University, West Lafayette, IN 47907, USA
[2] Department of Agronomy, Purdue University, West Lafayette, IN 47907, USA
*Correspondence to*: Qianlai Zhuang (qzhuang@purdue.edu)
**Abstract.** Northern peatlands are a large C sink during the Holocene, but whether they will keep being a C sink
under future climate change is uncertain. This study simulates the responses of northern peatlands to future climate until
2300 with a Peatland version Terrestrial Ecosystem Model (PTEM). The simulations are driven with two sets of CMIP5
climate data (IPSL-CM5A-LR and bcc-csm1-1) under three warming scenarios (RCP2.6, 4.5 and 8.5). Peatlands expansion,
shrink, accumulation and decomposition are modeled. In the 21$^{st}$ century, northern peatlands are projected to be a C source
of 1.2-13.3 Pg C under all climate scenarios except for RCP 2.6 of bcc-csm1-1 (a sink of 0.8 Pg C). During 2100-2300,
northern peatlands under all scenarios are a C source under IPSL-CM5A-LR scenarios, being larger sources than bcc-csm1-1
scenarios (5.9-118.3 vs. 0.7-87.6 Pg C). The peatland being C sources are due to: 1) water table depth (WTD) becomes
deeper and permafrost thaw increases decomposition rate; 2) net primary production (NPP) does not increase much as
climate warms because peat drying suppresses net N mineralization and 3) as WTD deepens, peatlands switches from moss-
herbaceous dominated to moss-woody dominated, while woody plants require more N for productivity. Under IPSL-CM5A-
LR scenarios, northern peatlands remain as a C sink until pan-Arctic annual temperature reaches -2.6 -  -2.89°C, while this
threshold is -2.09 -  -2.35°C under bcc-csm1-1 scenarios. This study predicts an earlier northern peatland sink to source shift
than previous estimates in the literature and emphasizes the vulnerability of northern peatlands to climate change.

## 21  1 Introduction

22       Peatlands are an ecosystem type that characteristically has more than 30cm peat thickness comprised of more than

30% organic materials within the peat layer. The formation of this thick organic soil layer requires wet and low oxygen
conditions that prevent dead plant litter from fully decomposed (Finlayson and Milton, 2018). Around 85% of global
peatlands C storage is in northern high latitude regions (415±150 Pg C) (Nichols and Peteet, 2019; Turunen et al., 2002)
where low temperature and relatively high precipitation create favorable conditions for peat accumulation (Xu et al., 2018;
Hugelius et al., 2020).

Peatlands are vulnerable to disturbances induced by climate warming (Loisel et al., 2021), especially when the

warming in the Arctic region is more severe than in other regions (Allen et al., 2018). First, warming influences northern
terrestrial ecosystem vegetation productivity by increasing spring photosynthesis and prolongs growing season (Piao et al.,





2008; Helbig et al., 2017). Second, warming could induce drier Arctic conditions (Finger Higgens et al., 2019), and
peatlands water table drawdown will result in net increase of greenhouse gas emissions (Huang et al., 2021). Third,
decomposition rate increases under higher temperature and previous studies found positive linear correlations between
warming and net C loss rate (Hanson et al., 2020). Fourth, permafrost thaw under warming conditions will expose
previously-frozen C for decomposition (Gandois et al., 2019). To date, multiple studies have explored northern peatland
responses to future climate changes (Loisel et al., 2021; Qiu et al., 2020; Chaudhary et al., 2020; Müller and Joos, 2021).
However, the projection of northern peatland C sink capacity during the 21st century is highly diverse including sink-to-
source switch (Chaudhary et al., 2017; Müller and Joos, 2021), higher sink capacity under mild climate changes (Qiu et al.,
2020), and the reduced C sink capacity (Chaudhary et al., 2020).

Given the uncertainties of northern peatlands response to future climate changes, modeling peatland C dynamics
considering peatland extent changes could improve the accuracy of future projection. In this study, a process-base model, the
Peatland Terrestrial Ecosystem Model (PTEM 2.2), is used to address this issue. PTEM 2.0 has been modified in terms of
plant functional type (PFT), peat accumulation and decomposition, fen-bog transition and soil thermal module to better
represent peatland ecosystem processes (Zhao et al., 2022b). The revised PTEM 2.0 is able to capture peat core age-depth
profile at site level (Zhao et al., 2022b) and has been further modified and applied to simulate Holocene (PTEM 2.1, 15ka
BP - 1990) pan-Arctic peatland accumulation and expansion at 0.5° resolution (Zhao et al., 2022a). The estimated pan-Arctic
peatland C stock is 396-421 Pg C and Holocene average C accumulation rate (CAR) is 22.9 g C·m-2 yr-1 (Zhao et al.,
2022a). The values and spatial pattern of soil C stock are in a close agreement with Qiu et al. (2019), Hugelius et al. (2020),
Spahni et al. (2013) and Hugelius et al. (2013), and the values and temporal pattern of CAR are consistent with Loisel et al.
(2014), Chaudhary et al. (2020) and Nichols and Peteet (2019). In this study, the results of the Holocene simulation are used
as the initial condition for the future simulation.

The methods used in Holocene simulation can not be applied directly to future simulation due to two issues. First,
previous studies on future peatland C dynamics are mostly based on fixed peatland extent (Loisel et al., 2021; Qiu et al.,
2020; Chaudhary et al., 2020; Müller and Joos, 2021). However, the future peatland extent is likely to vary under climate
change. To address this issue, we enhance PTEM 2.2 to simulate wetland dynamic extent during 1990-2300 at sub-grid cell
scales. Notably, although the spatially explicit peat expansion process was considered in the Holocene simulation, the sub-
grid cell expansion trend was simply derived from the fitted curve of existing pan-Arctic peat basal dates (Zhao et al.,
2022a). It is problematic to apply this fitted trend to future simulation since severe climate changes may interrupt the
Holocene peat expansion pattern. Therefore, a different approach of estimating peatland extent needs to be developed for
future simulation.

Second, in PTEM 2.2, peatland water table depth (WTD) and nutrient availability is influenced by run-on. Previous
PTEM 2.1 assumes run-on is a function of peat thickness and the theoretically maximum run-on when peat thickness is set to
0 cm. Under relatively stable climate conditions during the Holocene after peat initiation, the theoretically maximum run-on
is assumed to be a constant (i.e., parameter) and thereby run-on solely depends on peat thickness (Zhao et al., 2022a).





However, when climate becomes wetter or drier in the future, the theoretically maximum run-on could vary significantly and
the original PTEM 2.1 assumption becomes problematic. Therefore, it is necessary to revise the hydrology module of PTEM
2.2 such that run-on could respond to climate change.

To address these two issues, a TOPMODEL approach is used (Lu and Zhuang (2012). The TOPMODEL approach

downscales coarse grid cell WTD into finer resolutions given the sub-grid-cell topographic wetness index (TWI) and decay
parameter (*f*) (Beven and Kirkby, 1979). Previous studies have combined TEM, TOPMODEL and variable infiltration
capacity (VIC) model to estimate Alaska Yukon river basin methane emissions using 1km resolution WTD interpolated from
30km resolution (e.g., Lu and Zhuang (2012)). By applying TOPMODEL, we are able to estimate the dynamics of a) sub-
grid-cell WTD; b) the spatially explicit wetland fraction defined by annual WTD threshold (25cm, Fan et al. (2013)); c) sub
grid-cell peat accumulation and decomposition given interpolated WTD; and d) the spatially explicit peatland fraction
defined by peat thickness threshold (30cm, Finlayson and Milton (2018)). Furthermore, soil moisture can be estimated from
WTD, with which we can estimate run-on from the difference between interpolated WTD and simulated WTD without run-
on.

With peatland dynamics being simulated both horizontally (i.e., peatlands expansion and shrink) and vertically (i.e.,

peat accumulation and decomposition), this study aims to answer the following questions: a) how will the C sink of pan-
Arctic peatlands change during 1990-2300? b) What are the major drivers for these changes? c) How does the pan-Arctic
peatlands C sink respond to unit temperature and precipitation increase? and d) What is the threshold temperature and
precipitation for pan-Arctic peatland C sink and source shift?
**2 Methods**

In this study, two CMIP5 climate model products (IPSL-CM5A-LR and bcc-csm1-1) are selected as climate inputs,

with three warming scenarios considered (RCP 2.6, RCP 4.5 and RCP 8.5). The simulation is divided into two parts: 1)
simulating grid cell average WTD with PTEM 2.2 and interpolating grid cell WTD into sub-grid cell scale with the
TOPMODEL approach; and 2) simulating sub-grid cell scale peat accumulation and decomposition in current and potential
peatland regions (Figure 1). Although this study aims at the peatland dynamics after 1990, the simulations start in 1940. The
simulation during 1940-1990 works as spin up process and is also used for calibration against historical data.
**2.1 Selection of climate input data**

In the previous PTEM 2.0 site-level simulation, among many CMIP5 data products, IPSL-CM5A-LR product was

selected as climate input because it provides long temporal coverage (1850-2300) for RCP 2.6, RCP 4.5 and RCP 8.5
scenarios (Zhao et al., 2022b). In addition, it shows a good agreement with CRU temperature in Eurasia and low biases in
historical temperature and precipitation in North America (Miao et al., 2014; Sheffield et al., 2013). However, IPSL-CM5A-
LR product also shows more extreme warming than the other CMIP5 products, especially under RCP 8.5 (Palmer et al.,
2018). To address the uncertainty caused by climate inputs, another CMIP5 product, bcc-csm1-1, covering 1850-2300, three



RCP scenarios, projecting milder future climate warming, is selected. In order to run PTEM 2.2, temperature, precipitation,
cloudiness and vapor pressure are required. Neither of IPSL-CM5A-LR and bcc-csm1-1 model provides vapor pressure data,
which are thus calculated with temperature and relative humidity (Zhao et al., 2022b). Both climate products are bias-
corrected to CRU data (v4.03, Harris et al. (2014)) during 1940-1990 as described in Zhao et al. (2022b). The bias correction
makes sure that the difference in future simulations under two climate inputs are mostly introduced by the different level of
post-1990 warming, rather than the difference in their historical records before 1990.






**Figure 1. Flow chart of Method Section 2.2 and 2.3.**




## 2.2 Future grid cell average WTD simulation

### 2.2.1 TOPMODEL parameter estimation

In this study, the peatland condition in 1940 derived from previous Holocene simulation is used as the initial condition for future peatland simulations (1940-2300) (Zhao et al., 2022a). In order to be consistent with the Holocene simulation, before running future WTD simulation, it is necessary to make sure that interpolating the PTEM-simulated recent WTD by TOPMODEL could derive the wetland extent as shown in the end of the Holocene simulation. In particular, 'recent' in this study refers to 1940-1990, and wetlands are defined as the region with long-term annual WTD shallower than 25cm (Fan et al., 2013). To satisfy this requirement, the TOPMODEL parameters need to be estimated before WTD simulation. TOPMODEL describes sub-grid cell WTD variation with topography. The topography effects on the local WTD are estimated with topographic wetness index (TWI) values, the larger TWI values indicate the shallower WTD and higher flooding probability (Stocker et al., 2014). In order to estimate sub-grid cell wetland and peatland conditions at 1% accuracy, each $0.5°×0.5°$ grid cell is divided into 100 bins by the TWI histogram (Figure 1 (a)). With global terrestrial TWI values available at 15 arcsec resolution (Marthews et al., 2015), each bin is composed of 36 TWI values where water bodies have null values. For bin $i$ within a given grid cell:

$$z_{wti} = z_{wt} - \frac{1}{f} \times (k_i - \lambda) \tag{1}$$

Where $z_{wti}$ is the WTD of bin $i$, $k_i$ is the average TWI of bin $i$, $\lambda$ is the grid cell average TWI, $z_{wt}$ is the grid cell average WTD and $f$ is the decay parameter. In Eq. (1), the parameters need to be estimated are $z_{wt}$ and $f$. In particular, the $z_{wt}$ here refers to the 50-year average WTD during 1940-1990. $z_{wt}$ and $f$ values are calculated as:

$$\begin{cases} f = 2.6 & \text{(2a)} \\ z_{wt-thres} = z_{wt} + \frac{1}{f}(\lambda_{thres} - \lambda) & \text{(2b)} \end{cases}$$

Where $f$ value is from Kleinen et al. (2020). $z_{wt-thres}$ is the threshold WTD of wetlands (i.e., -0.25m) and $\lambda_{thres}$ is the TWI value corresponding to $z_{wt-thres}$. For a given grid cell where wetland abundance is n% (n is an integer), $\lambda_{thres}$ is the TWI value of the n-th largest TWI values among 100 bins. The spatially explicit wetland fraction (n%) during 1940-1990 is consistent with the wetland fraction in 1990 in the Holocene simulation, which is the average value of three peatland maps covering the pan-Arctic region (Xu et al., 2018; Hugelius et al., 2020; Melton et al., 2022). The shallowest WTD in each grid cell is:

$$z_{wt-max} = z_{wt} + \frac{1}{f}(\lambda_{max} - \lambda) \tag{3}$$

Where $\lambda_{max}$ is the maximum TWI value in 100 bins. If $z_{wt-max}$ is greater than zero (i.e., above surface), $z_{wt-max}$ is assumed to be -0.01m, and $z_{wt}$ and $f$ values are calculated by Eq. (2b) and Eq. (3). Otherwise, $z_{wt}$ and $f$ values are calculated by Eq. (2a) and Eq. (2b).

### 2.2.2 PTEM revisions





The PTEM 2.1 used in pan-Arctic Holocene simulations is able to estimate the wetland WTD in a given grid cell,
while not the grid-cell average WTD composed of both wetland and non-wetland land covers (Zhao et al., 2022a). In order
to simulate the grid-cell average WTD, PTEM 2.1 is revised by applying some of the algorithms from VIC model. VIC
model was developed by Liang et al. (1994) and has been updated to version 5 (VIC-5, Hamman et al. (2018)). Compared
with the hydrology module of PTEM 2.1, VIC has the same soil vertical structure of three layers. The major hydrological
processes including canopy interception of precipitation, infiltration, gravity-driven vertical flow, evapotranspiration, upper
soil layer evaporation, effect of frozen-thaw on soil moisture are considered in both models (Liang et al., 1994; Zhuang et
al., 2002). With similar structure, processes and variables, it is possible to apply some of VIC algorithms to PTEM 2.1. In
particular, the algorithms of surface runoff, vertical flow from upper to lower layers, the computation of base flow and the
estimation of WTD from given soil moisture are added to PTEM 2.1. The computation of surface runoff in VIC is based on
the Xinanjiang model that assumes runoff is the amount of precipitation that falls on the saturated fraction of a grid cell
(Zhao et al., 1980). The relationship between soil water storage and saturated fraction is given by:
$i = i_m[1 - (1 - A)^{1/B}]$                                                                                  (4)
Where A is the fraction of the grid cell that the infiltration capacity (i.e., the possible maximum depth of water
stored in soil column given area fraction) is less than $i$, $i_m$ is the maximum infiltration capacity within the given grid cell,
and B is a shape parameter (Wood et al., 1992). The calculation of the uppermost layer runoff given precipitation and the
initial soil moisture is well documented in Wood et al. (1992) (Eq. (1-3)) and Liang et al. (1994) (Eq. (13), Eq. (17-18)). In
addition, gravity-driven water flow from upper to lower layers is given by Liang et al. (1994) (Eq. (18-20)) based on upper
layer soil moisture, residual moisture content, pore size distribution index and the hydraulic conductivity estimated from
Brooks (1965). Following VIC, PTEM 2.2 also assumes base flow only happens in the bottom soil layer. The computation of
base flow in VIC is derived from the model in Franchini and Pacciani (1991) and the equations are listed in Liang et al.
(1994) (Eq. (21)). Computing WTD given soil moisture was first used in VIC 4.1.2 (Bohn et al., 2013). Edited from VIC-5,
the WTD-soil moisture relationship in PTEM 2.2 is given by:
$W_{tot} = W_{avg} \times (SM_{max} - SM_{res}) + SM_{res}$                                                     (5)
Where $W_{tot}$ is the total soil moisture of three layers (mm), $W_{avg}$ is the average relative soil moisture, $SM_{max}$ is the
maximum soil moisture (mm) and $SM_{res}$ is the residual soil moisture (mm). With $SM_{max}$ and $SM_{res}$ being spatially explicit
parameters, $W_{avg}$ is:
$W_{avg} = (D_{tot} - z_{wt} - \frac{b}{b-1} \times bubble \times (1 - \left(\frac{z_{wt}+bubble}{bubble}\right)^{\frac{b-1}{b}}))/D_{tot}$     (6)
Where $D_{tot}$ is the total depth of soil layer (cm), $z_{wt}$ is given WTD (cm below surface), $bubble$ is the bubbling
pressure (cm), $b$ is the parameter:
$b = 0.5 \times (expt - 3)$                                                                                   (7)
Where $expt$ is the exponent parameter from Brooks-Corey relationship, and is always greater than 3 (Rawls et al.,
1992). In PTEM 2.2, the spatially explicit relationship between WTD and total soil moisture is given by Eq. (5-7) at 5cm



WTD interval. During simulation, PTEM 2.2 calculates the total soil moisture and finds the corresponding WTD. In case of
soil moisture does not correspond with any 5-cm interval WTD, PTEM 2.2 will find the closest upper and lower soil
moisture values in the soil moisture-WTD profile and interpolate from the upper and lower WTD values.
In site-level and Holocene simulations, there are three PFTs in PTEM 2.0 and 2.1: moss, herbaceous plant and
shrub (Zhao et al., 2022b). However, trees are also an important PFT in northern peatlands (Hanson et al., 2020). Therefore,
in both grid-cell average WTD and sub-grid cell peatland simulations, it is necessary to include trees as a PFT. In particular,
the vegetation C and N pool in PTEM 2.2 are now divided into four sub-pools: moss, herbaceous plant, shrub and tree. The
dominance of these four PFTs are determined by WTD and their maximum possible productivity. The litter fall from four
PFTs becomes the input of soil C and N, and the decomposition ability of litter is influenced by the fraction of litter origin
from each PFT. The calculations of C and N cycles of trees are the same as the other three PFTs, although controlled by
different PFT-specific parameters. The detailed description and equations are documented in Zhao et al. (2022b).
The calculation of evapotranspiration (EET) in PTEM 2.2 is derived from FAO algorithm for calculating crop EET
(Allen et al., 1998):
$$EET = PET \times k_c \times foliage + E_{soil} \times (1 - foliage) \qquad (8)$$
Where PET is the potential evapotranspiration given by Penman-Monteith model in PTEM 2.2, $k_c$ is a coefficient,
$E_{soil}$ is the evaporation from bare land and $foliage$ is a PTEM 2.2 variable describing the relative abundance of leaf
biomass (0-1). Although the FAO algorithm is widely applied in estimating crop EET, it is also proved applicable to
shrubland, grassland and forest (Liu et al., 2017). In PTEM 2.2, $k_c$ is calculated as:
$$k_c = \sum_{i=1}^{3} k_{c-pft} \times w_{pft} \qquad (9)$$
Where three vascular PFTs are considered influential to EET (i.e., herbaceous plant, shrub and tree), $k_{c-pft}$ is the
spatially explicit coefficient for given PFT, and $w_{pft}$ is the weight of given PFT estimated from its dominance:
$$w_{pft} = \frac{VEGC_{pft}}{\sum_{i=1}^{3} VEGC_{pft}} \qquad (10)$$
Where $VEGC_{pft}$ is the vegetation C of given PFT, and only three vascular PFTs are used for weight calculation. In
WTD simulation, we assume no run-on from adjacent grid cells, thereby the grid cell water balance is:
$$\Delta SM = P - R_{off} - B - EET \qquad (11)$$
Where $\Delta SM$ is the change of soil moisture, $P$ is precipitation, $R_{off}$ is surface run-off and $B$ is the bottom layer base
flow.
**2.2.3 Grid cell average WTD simulation and post-processing**
Adding VIC algorithms to PTEM 2.2 requires VIC parameters at 0.5° resolution. These parameters include variable
infiltration curve parameter (binfilt), maximum velocity of base flow (Dsmax), fraction of Dsmax where non-linear base
flow begins (Ds), fraction of maximum soil moisture where non-linear base flow occurs (Ws), exponent used in base flow
curve (c), expt in Eq. (7), saturated hydrologic conductivity (Ksat), depth of three soil layers (depth), bubbling pressure of
soil layers (bubble), bulk density of soil layers (bulk_density) and soil particle density (soil_density). These parameter values





are available globally at $(1/16)°$ resolution (Schaperow and Li, 2021) and are aggregated into 0.5° resolution in this study. To
run PTEM 2.2, in addition to the climate inputs, the historical (1940-1990) $CO_2$ concentration (ppm) is derived from TraCE
21ka dataset (He, 2011). The $CO_2$ concentration for three RCP scenarios (1991-2300) is provided by Meinshausen et al.
(2011). Spatially explicit soil texture (Fao/Unesco, 1974) and elevation (Zhuang et al., 2002) were also required.
Before conducting WTD simulation, spatially explicit calibration for annual PET and $k_{c-pft}$ are conducted.
Spatially explicit calibration for annual PET is conducted because the original PTEM 2.1 parameters estimate unreasonably
large PET. Therefore, the global aridity index and potential evapo-transpiration (ET0) database v3 (Zomer and Trabucco,
2022) is selected as a reference. The dataset is selected because its annual PET is the long-term value of 1970-2000, which
can be the approximate reference to the PET during 1940-1990 in this study. In addition, the reference dataset is also based
on Penman-Monteith model but with more detailed estimation on the parameters than PTEM (Zomer and Trabucco, 2022).
The 30 arcsec resolution reference PET is aggregated into 0.5° resolution for calibration. The spatially explicit Penman-
Monteith parameters in PTEM 2.2 are calibrated with PEST (v17.2 for Linux). Since both reference dataset and PTEM 2.2
estimate PET with the same model, the calibration result is close to the reference for both IPSL-CM5A-LR and bcc-csm1-1
climate inputs (SI Figure 1).
After PET calibration, the spatially explicit calibration for $k_{c-pft}$ is conducted such that the 50-year WTD is
consistent with the $z_{wt}$ calculated by Eq. (2-3). Same as PET calibration, spatially explicit $k_{c-pft}$ values are also calibrated
by PEST (v17.2 for Linux). The wetland abundance in the end of the Holocene simulation (i.e., reference dataset) (Xu et al.,
2018; Hugelius et al., 2020; Melton et al., 2022) and the wetland abundance interpolated by TOPMODEL from calibrated
WTD (average of 1940-1990) is shown in SI Figure 2. Since each grid cell is divided into 100 bins by TWI values, the
minimum wetland abundance is 1%. In this study, the grid cells with less than 1% wetland are not used for peat simulation.
Leaving out the grid cells with less than 1% wetlands, the pan-Arctic wetlands area for the reference dataset is 2.93 Mkm$^2$,
the calibrated wetlands area with IPSL-CM5A-LR forcing input is 2.81 million km$^2$, and with bcc-csm1-1 forcing input is
2.86 million km$^2$.
After calibration, the WTD simulation is conducted for 1940-2300 at 0.5° resolution (Figure 1 (a)). Notably, WTD
simulation only aims at estimating grid cell average WTD and the peat accumulation and decomposition processes are not
simulated. The grid cell average WTD during 1940-2300 is interpolated by TOPMODEL using the parameters calculated in
Section 2.2.1. The changes of wetlands extent during 1990-2300 under IPSL-CM5A-LR and bcc-csm1-1 forcing inputs are
presented in SI Figure 3 and 4.
**2.3 Peatland simulation**
**2.3.1 PTEM revision**
The TOPMODEL-interpolated bin WTD is used as an input in peatland simulation (Figure 1 (b)). In contrast to the
WTD simulation where the grid cell run-on is assumed to be zero (Eq. (11)), the run-on in peatland simulation is calculated
with a water balance equation:





$\quad \Delta SM = P + R_{on} - R_{off} - B - EET$ (12)
$\qquad$ Where $\Delta SM$ is the difference between soil moisture at two adjacent time steps (i.e., months), and the soil moisture
in each month is estimated form the input WTD and the WTD-soil moisture relationship given by Eq. (5-7). The run-off
($R_{off}$), base flow ($B$) and evapotranspiration ($EET$) are calculated in the same way as in WTD simulation. In the Holocene
simulation, soil pH value is calculated as a function of run-on which is solely controlled by peat thickness. In the revision,
soil pH is calculated as:
$\quad pH = -log_{10}(n^{H^+}/SM)$ (13)
$\qquad$ Where $pH$ is the soil pH value, $n^{H^+}$ is the number of $H^+$ particles, and $SM$ is the soil moisture (mm). Notably, on
unit area (i.e., 1m$^2$), 1mm soil moisture is equal to 1L soil water. Therefore, $n^{H^+}/SM$ calculates the concentration of $H^+$
particles per liter. And the number of $H^+$ particles is calculated as:
$\quad \Delta H^+ = 10^{-pH_p} \times P + 10^{-pH_{ron}} \times R_{on} - 10^{-pH_w} \times EET - 10^{-pH_0} \times (R_{off} + B)$ (14)
$\qquad$ Where $pH_p$ is the pH value of precipitation (assumed 5.0), $pH_{ron}$ is the pH value of run-on water (assumed 7.0),
$pH_w$ is the pH value of EET water (assumed 7.0), and $pH_0$ is the pH value of soil water at previous month. The spatially
explicit initial pH values are from (Carter and Scholes, 2000).
$\qquad$ In Holocene simulations, CH$_4$ production is simulated, but since oxidation process is not considered, CH$_4$ emission
is not calculated. In this revision, CH$_4$ oxidation is enabled and thereby it is possible to estimate net CH$_4$ emission. The
algorithms are documented in Zhuang et al. (2004).
**2.3.2 PTEM simulation**
$\qquad$ In each grid cell, among the 100 bins, the bins that the long-term WTD has ever been shallower than 25cm are
classified as 'potential peatlands', which are used for peatland simulation (Figure 1 (b)). To be consistent with the WTD
simulation, long-term WTD refers to the 50-year moving average of annual WTD. In this study, we assume within each grid
cell, the climate conditions are similar and the key control of whether peat exists at sub-grid cell scale is the local WTD
influenced by sub-grid topography. Therefore, for all the bins in the same 0.5°×0.5° grid cell, the forcing data, soil texture,
elevation and parameters are the same except for the input WTD.
$\qquad$ In Holocene simulations, the maximum C assimilated by ecosystem parameter ($c_{max}$) is calibrated for over 2000
peat cores and interpolated into the pan-Arctic region (Zhao et al., 2022a). The calibration process reduces the uncertainty
from forcing data, other parameters and model structure, and the simulated spatial and temporal pattern of pan-Arctic
peatland C stock is consistent with multiple datasets (Zhao et al., 2022a). However, since the hydrology module of PTEM
2.2 is revised and peat accumulation and decomposition is sensitive to hydrological processes, using the original parameters
could result in considerable bias. In order to make sure the revised PTEM 2.2 simulates consistent C accumulation rate
(CAR) with the previous study, a spatially explicit calibration on maximum C assimilated by ecosystem ($c_{max}$) parameter is
conducted.



Before calibrating CAR, it is necessary to initialize PTEM 2.2 with reasonable peat conditions. To initialize the
simulation, the peat profile in 1940 derived from the Holocene simulation (Zhao et al., 2022a) is used (Figure 1 (b)). In
particular, the peat profile records the physical property of vertical peat layers including bulk density, organic C density,
layer thickness (1cm except for the top layer), fraction of remaining undecomposed organic matter and decomposition rate of
undecomposed organic matter at 0°C. This information can be used to estimate the decomposition rate of existing peat given
WTD, soil pH and soil temperature (Zhao et al., 2022b).
With initial peat profile as an input, $c_{max}$ values are calibrated with PEST (v17.2 for Linux) (Figure 1 (b)). In
particular, within each grid cell, the 50-year average CAR of historical wetland bins (i.e., the bins that are classified as
wetlands during 1940-1990) are simulated and averaged to get the grid cell average 50-year peatland CAR. This grid cell
average peatland CAR is calibrated against the CAR derived from the Holocene simulation during the same period (SI
Figure 5).
After calibration, the peat simulation is conducted for all pan-Arctic potential peatland bins during 1940-2300
(Figure 1 (b)). For the Greenland grid cells not included in the Holocene simulation and thereby have no calibrated $c_{max}$
values, the $c_{max}$ values are interpolated from adjacent grid cells. For the bins not included in the Holocene simulation or not
being peatlands before 1990, the peat profile is initialized as 3cm fully decomposed peat. Notably, under different forcing
data and warming scenarios, the number and distribution of potential peatland bins are slightly different, which makes the
initial pan-Arctic peatland C storage in 1940 slightly different (SI Table 1). When running peat simulation, the forcing input
(temperature, precipitation, cloudiness and vapor pressure), soil texture, elevation and parameters are the same as the ones
used in the WTD simulation, except for the spatial-explicit $c_{max}$ values.
**2.3.3. Peat simulation post-processing**
After simulation, the simulated results are analyzed in terms of 1) the temporal pattern of pan-Arctic climate
dynamics; 2) the temporal pattern of pan-Arctic peatland C stocks and C fluxes; 3) the main drivers of pan-Arctic peatland C
dynamics; and 4) the threshold temperature and precipitation of pan-Arctic C sink/source shift.
Threshold temperature is calculated with logistic regression:
$$f(temp) = \begin{cases} 0 & NEP \leq 0 \\ 1 & NEP > 0 \end{cases} \qquad\qquad (15)$$
Where NEP is net ecosystem productivity. A fitting curve of $f(temp)$ is derived for the pan-Arctic region and for
each grid cell. Under sink-source shift, the fitting curve rises from 0 to 1, and the threshold temperature of sink-source shift
is determined when $f(temp)$ is 0.5. The threshold precipitation is calculated in the same way.
**3. Results**
**3.1 Warmer and drier pan-Arctic peatlands during 1990-2300**





**Figure 2. Time series of pan-Arctic annual air temperature (°C), annual precipitation (mm·yr⁻¹), wetland area (Mkm²) and permafrost area in peatland regions (Mkm²) during 1990-2300.**

Both IPSL-CM5A-LR and bcc-csm1-1 climates show higher temperature and precipitation during 1990-2300. In particular, under RCP 2.6 and RCP 4.5, temperature increases mostly before 2100 by 2.3-4.1°C and 2.0-3.2°C under IPSL-CM5A-LR and bcc-csm1-1, respectively (Figure 2 1(a-b), SI Table 2). Meanwhile, precipitation increases by 40.7-59.7 mm·yr⁻¹ and 38.1-53.9 mm·yr⁻¹ for under IPSL-CM5A-LR and bcc-csm1-1 (Figure 2 2(a-b), SI Table 2). During 2100-2300, under RCP 2.6, the temperature decreases by 0.8°C in IPSL-CM5A-LR and by 1.1°C in bcc-csm1-1. Under RCP 4.5, temperature keeps increasing but in a slower rate than before 2100 (IPSL-CM5A-LR: 1.3°C vs. bcc-csm1-1: 0.4°C). Meanwhile, precipitation slightly decreases under RCP 2.6 (IPSL-CM5A-LR: -12.0 mm·yr⁻¹ vs. bcc-csm1-1: -5.8 mm·yr⁻¹) while increases under RCP 4.5 (IPSL-CM5A-LR: 29.1 mm·yr⁻¹ vs. bcc-csm1-1: 8.5 mm·yr⁻¹) (Figure 2 1(a-b) &2(a-b), SI Table 2). Different from RCP 2.6 and RCP 4.5, the increase in temperature and precipitation under RCP 8.5 is stable throughout 1990-2300. In particular, under IPSL-CM5A-LR, during 1990-2100 and 2100-2300, temperature increases by 8.4





and 8.1 °C while precipitation increases by 106.1 and 131.7 mm·yr⁻¹. Under bcc-csm1-1, during 1990-2100 and 2100-2300,
temperature increases by 7.2 and 6.9 °C while precipitation increases by 100.9 and 198 mm·yr⁻¹, respectively (Figure 2 (1-
2)c, SI Table 2).

The result of pan-Arctic wetland shrinking under all scenarios indicates that the increase of precipitation does not

compensate the increase of evapotranspiration under warmer climate. Therefore, the pan-Arctic generally becomes drier and
WTD becomes deeper (Figure 2 3(a-c)). In particular, during 1990-2100, under IPSL-CM5A-LR, wetland shrinks by 0.6,
0.9 and 1.4 million km² under three RCP scenarios. Meanwhile, under bcc-csm1-1, wetland shrinks slightly less by 0.4, 0.6
and 1.2 million km² under three RCP scenarios, respectively. During 2100-2300, under both IPSL-CM5A-LR and bcc-csm1-
1, wetlands slightly expand by 0.1 million km² under RCP 2.6, while under the warmer scenarios, wetland further shrinks by
0.2 and 0.9 million km², respectively (Figure 2 3(a-c), SI Table 2).

Following climate warming, permafrost shrink is simulated across the current pan-Arctic peatland region under all

scenarios (Figure 2 4(a-c)). In particular, with IPSL-CM5A-LR forcing, under RCP 2.6, 4.5 and 8.5, permafrost shrinks by
0.2, 0.7 and 1.2 million km² during 1990-2100 and expands by 0.1, shrinks by 0.1 and 0.5 million km², respectively, during
2100-2300. Meanwhile, active layer deepening is simulated in the remaining permafrost region (SI Figure 6). Similarly, with
bcc-csm1-1 forcing, under RCP 2.6, 4.5 and 8.5, permafrost shrinks by 0.2, 0.2 and 1.0 million km² during 1990-2100 and
expands by 0.1, shrinks by 0.1 and 0.6 million km², respectively, during 2100-2300 (Figure 2 4(a-c), SI Table 2).
Meanwhile, active layer deepening is simulated in the remaining permafrost region under RCP 4.5 and 8.5 (SI Figure 7).

Under RCP 2.6 and 4.5, with both IPSL-CM5A-LR and bcc-csm1-1 forcing, peatlands (i.e., the region with peat

thickness >= 30cm) area expands during 1990-2300 (Figure 3). In particular, the new peat area expands by 0.1 to 0.2 million
km², while the old peat area is stable (SI Table 2). Under RCP 8.5, however, peatland area shrinks. In particular, although
new peat land area expands by 0.1 million km², the old peatland area shrinks by 0.1 to 0.4 million km², causing total peatland
area decrease (Figure 3, SI Table 2).



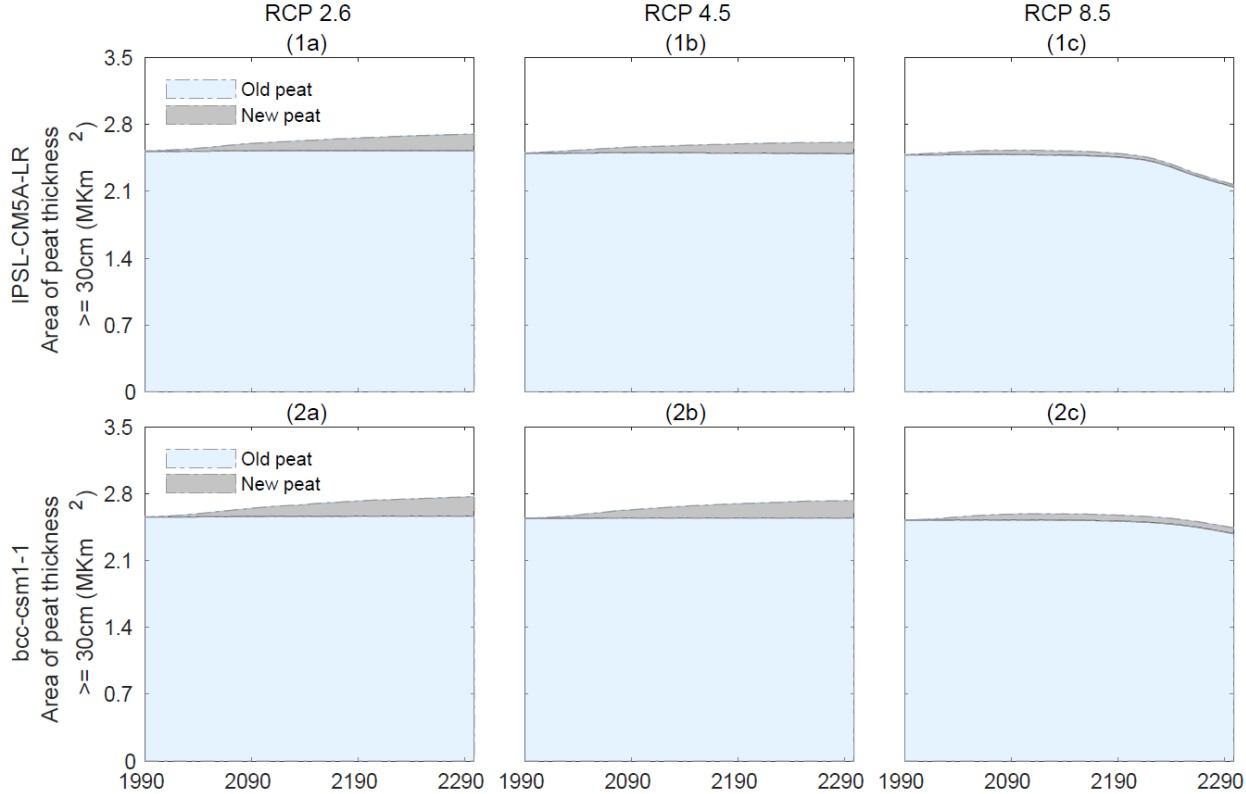

**Figure 3. Time series of pan-Arctic old and new peatland area (million km$^2$) during 1990-2300.**

## 3.2 Pan-Arctic C stocks and fluxes under climate change

### 3.2.1 Before 2100

With WTD becomes deeper, active layer depth (ALD) becomes deeper and permafrost extent shrink, it is reasonable that decomposition increases during 1990-2100 under all scenarios (Figure 4 3&4 (a-c), SI Table 3). Meanwhile, NPP slightly decreases with IPSL-CM5A-LR forcing while increases with bcc-csm1-1 forcing (Figure 4 2(a-c), SI Table 3). In PTEM 2.2, NPP is primarily influenced by temperature and nitrogen availability, and available nitrogen mainly comes from net N mineralization. In all scenarios, net N mineralization rate increases (negative values indicate higher net N mineralization) during 1990-2100 (SI Figure 8), indicating more available N for vegetation. The increase in both N availability and temperature can not explain the reason for NPP decrease. However, NPP decrease can be explained by the shift in PFTs. In particular, during 1990-2100, with water table becomes deeper, the dominance of herbaceous plants is gradually replaced by woody plants (i.e., shrubs and trees) that can thrive under drier conditions (SI Figure 9). In PTEM 2.2, compared with herbaceous plants, woody plants require more nitrogen for production. Therefore, although N availability increases, the increase is not sufficient for woody plants to maintain as high NPP as herbaceous plants and the overall NPP decreases.





**Figure 4. Time series of pan-Arctic peatland C storage (vegetation and soil, Pg C), NPP (TgC·yr⁻¹), CO2 emissions (TgC·yr⁻¹) and CH4 emissions (TgC·yr⁻¹) during 1990-2300.**

In all scenarios except for bcc-csm1-1 RCP 2.6, the increase in decomposition overrides the increase in NPP and thereby C stock decreases (Figure 4 1(a-c), SI Table 3). In particular, with ISP-CM5A-LR forcing, by 2100, C stock decreases by 1.3, 5.2 and 13.3 Pg C under RCP 2.6, 4.5 and 8.5, respectively. With bcc-csm1-1 forcing, by 2100, C stock increases by 0.8 Pg C under RCP 2.6, while decreases by 1.2 and 7.8 Pg C under RCP 4.5 and 8.5, respectively (Figure 4 1(a-c), SI Table 3). Notably, although pan-Arctic peatlands are C sinks during 1990-2100 under bcc-csm1-1 RCP 2.6, the



sink is much lower than that during 1940-1990 with CAR decreases by 29.1 gC·m⁻²·yr⁻¹. Furthermore, this difference is
larger in the other scenarios (IPSL-CM5A-LR: 35.5-63.5 gC·m⁻²·yr⁻¹, bcc-csm1-1: 34.6-50.0 gC·m⁻²·yr⁻¹) (SI Figure 10&11,
SI Table 4).
**3.2.2 During 2100-2300**

During 2100-2300, the decrease in decomposition rate is simulated in RCP 2.6 and 4.5 with both forcing, while

decomposition rate becomes higher under RCP 8.5 (SI Table 3). Under RCP 2.6, the decrease in decomposition is driven by
the colder and wetter climate (Figure 2), while with IPSL-CM5A-LR forcing the decrease of C stock also influences
decomposition rate negatively. In contrast, under RCP 4.5 where climate becomes warmer and drier, the decrease in
decomposition rate is mostly driven by the lower C stock available for decomposition. However, under RCP 8.5 where the
climate change is more severe, the positive effect of warming and drying overrides the negative effect of insufficient C stock
and thereby decomposition rate keeps increasing (Figure 4 3&4(a-c), SI Table 3).

During 2100-2300, NPP in all scenarios decrease except for bcc-csm1-1 RCP 4.5 (Figure 4 2(a-b), SI Table 3). For

both forcings, under RCP 2.6 and 4.5, PFT distribution is stable after 2100 (SI Figure 9 1&2(a-b)). Therefore, NPP is driven
by the balance of net N mineralization and temperature. For bcc-csm1-1 RCP 4.5, the positive effect of temperature
overrides the negative effect of decreasing net N mineralization, while the opposite is found in the other scenarios (SI Figure
8). Under RCP 8.5, with further herbaceous-woody switch and decrease in net N mineralization, NPP decreases with both
forcings (Figure 4 2(c), SI Table 3). With NPP decrease and decomposition increase, pan-Arctic peatlands are C sources
under all scenarios. In particular, with IPSL-CM5A-LR forcing, under RCP 2.6, 4.5 and 8.5, pan-Arctic peatlands are
sources of 5.9, 22.5 and 118.3 Pg C, respectively, while these values are 0.7, 12.6 and 87.6 Pg C with bcc-csm1-1 forcing,
respectively (Figure 4 1(a-c), SI Table 3). During 2100-2300, CAR is lower than that during the 21st century. In particular,
under RCP 2.6, 4.5 and 8.5, CAR further decreases by 4.6-9.3 gC·m⁻²·yr⁻¹, 23.9-25.7 gC·m⁻²·yr⁻¹ and 135.5-193.8 gC·m⁻²·yr⁻¹
with IPSL-CM5A-LR forcing and 4.1-4.9 gC·m⁻²·yr⁻¹, 15.7-20.6 gC·m⁻²·yr⁻¹ and 103.7-145.3 gC·m⁻²·yr⁻¹ with bcc-csm1-1
forcing, respectively (SI Figure 10&11, SI Table 4).
**3.3 Pan-Arctic peatlands C sinks in response to climate change**

For both IPSL-CM5A-LR and bcc-csm1-1 forcings, the positive correlation between temperature and precipitation

is found at the pan-Arctic scale (Table 1). In particular, with 1°C annual temperature increase, the annual precipitation
increases by 13.84-15.33 mm·yr⁻¹ in IPSL-CM5A-LR forcing and 13.78-14.59 mm·yr⁻¹ in bcc-csm1-1 forcing. The
correlation has higher $R^2$ values in warmer scenarios. The positive correlation between temperature and precipitation is
mostly found in Eurasia and northeast America, where the $R^2$ values are also higher than the other region (SI Figure 12&13).

The negative correlation between temperature and pan-Arctic peatland C sink activity is found in both forcing

scenarios (Table 1). In particular, with 1°C annual temperature increase, pan-Arctic peatland C sink decreases by 40.46-
46.91 Tg C·yr⁻¹ in IPSL-CM5A-LR forcing and 33.27-41.1 Tg C·yr⁻¹ in bcc-csm1-1 forcing. The negative effect of
temperature is weaker in western Eurasia and Alaska regions, while stronger in the other regions where most of the current
peatlands exist (SI Figure 14&15). Due to the close positive correlation between temperature and precipitation, the

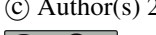



correlation between precipitation and pan-Arctic peatland C sink is also negative. In particular, with 1mm annual
precipitation increase, pan-Arctic peatland C sink decreases by 2.32-3.28 Tg C·yr[-1] in IPSL-CM5A-LR forcing and 1.85-2.92
Tg C·yr[-1] in bcc-csm1-1 forcing (Table 1). The spatial pattern of precipitation-C sink correlation is consistent with the spatial
pattern of temperature-C sink correlation (SI Figure 16&17).
At the pan-Arctic scale, a threshold annual temperature and precipitation can be found when peatlands switch from
a C sink to a source. In particular, with IPSL-CM5A-LR forcing, the threshold annual temperature is -2.89 - -2.6°C and the
threshold precipitation is 479.59 - 482.55 mm. With bcc-csm1-1 forcing, the threshold annual temperature is -2.35 - -2.09°C
and the threshold precipitation is 484.69 - 489.02 mm (Table 1). The threshold temperature varies spatially with mostly
below -3°C in the northern North American and western Eurasia regions and mostly above 1°C in the lower latitude regions
(SI Figure 18). Notably, the regions with below -3°C threshold temperature tend to have higher $R^2$ values (SI Figure 19). The
spatial pattern of precipitation threshold is consistent with temperature threshold and the region with 300-500mm annual
precipitation threshold has higher $R^2$ values, mostly seen in northern North American and western Eurasia (SI Figure
20&21).
**Table 1.** Relationship between pan-Arctic temperature, precipitation and C sink

| Model | RCP 2.6 | $R^2$ | RCP 4.5 | $R^2$ | RCP 8.5 | $R^2$ |
|---|---|---|---|---|---|---|
| Pan-Arctic peatlands C sink capability increases (TgC·yr[-1]) in response to 1°C annual temperature increase | | | | | | |
| IPSL-CM5A-LR | -43.92 | 0.72 | -40.46 | 0.86 | -46.91 | 0.96 |
| bcc-csm1-1 | -34.59 | 0.51 | -33.27 | 0.76 | -41.1 | 0.96 |
| | | | | | | |
| Pan-Arctic peatlands C sink capability increases (TgC·yr[-1]) in response to 1mm annual precipitation increase | | | | | | |
| IPSL-CM5A-LR | -2.32 | 0.64 | -2.47 | 0.78 | -3.28 | 0.92 |
| bcc-csm1-1 | -1.85 | 0.46 | -2.06 | 0.73 | -2.92 | 0.94 |
| | | | | | | |
| Annual temperature threshold of C sink-source conversion | | | | | | |
| IPSL-CM5A-LR | -2.89 | 0.57 | -2.72 | 0.9 | -2.6 | 0.86 |
| bcc-csm1-1 | -2.35 | 0.16 | -2.12 | 0.64 | -2.09 | 0.81 |
| | | | | | | |
| Annual precipitation threshold of C sink-source conversion | | | | | | |
| IPSL-CM5A-LR | 479.59 | 0.51 | 482.42 | 0.86 | 482.55 | 0.84 |
| bcc-csm1-1 | 489.02 | 0.16 | 485.5 | 0.63 | 484.69 | 0.79 |
| | | | | | | |
| Annual precipitation increase (mm) in response to 1°C annual temperature increase | | | | | | |
| IPSL-CM5A-LR | 15.33 | 0.74 | 14.71 | 0.9 | 13.84 | 0.98 |
| bcc-csm1-1 | 13.78 | 0.61 | 14.59 | 0.85 | 13.78 | 0.98 |


**4. Discussion**
**4.1 Wetlands and permafrost dynamics under climate change**





Wetlands loss is closely related to climate change and human activities. In particular, the loss has been found
globally since 1700AD, with 64-71% loss since 1900 AD (Davidson, 2014). Similarly, a more recent study has found 33%
of the global wetland loss as of 2009, with 45% in Europe and 8% in North America (Hu et al., 2017). In addition, regional
studies also report different scales of wetlands loss in China and coastal regions (Li et al., 2018; Niu et al., 2012). To date,
no many studies focus on future wetland extent simulations and the inconsistency among current wetland extent datasets
exists (Loveland et al., 2000; Friedl et al., 2002; Lehner and Döll, 2004; Bartholomé and Belward, 2005). Similar to this
study, one study highlighted the vulnerability of Arctic wetland extent in the 21st century due to permafrost thaw, although
most of the permafrost Arctic wetlands can remain stable under RCP 2.6 until at least 2100 (Kåresdotter et al., 2021).
The active layer depth (ALD) simulated by PTEM 2.2 is compared with two datasets derived from satellite data and
models, covering pan-Arctic region and Alaska, respectively (Obu et al., 2020; Yi and Kimball, 2020). The correlation with
pan-Arctic dataset (2001-2018) is higher than the correlation with Alaska dataset, while the overall estimation is consistent
between our study and two regional datasets (SI Table 3&4). Consistent with our study, Smith et al. (2022) found deepening
ALD since the 1990s in the permafrost region, indicating permafrost thaw could continue in warmer future and possibly in a
higher rate. The permafrost thaw progress in the 21$^{st}$ century agrees with the dynamics simulated by CCSM4 model,
suggesting that the CCSM4 permafrost area shrinks by 64% by 2100 under RCP 8.5 compared to our estimation of 53-60%
in this study (Lawrence et al., 2012).

**4.2 Future productivity and decomposition in northern peatlands**

In this study, NPP does not always increase under warmer climate due to PFT switch and net N mineralization rate
limiting. The overall trend of pan-Arctic peatland PFT switch is the expansion of woody plants and shrink of herbaceous
plants (SI Figure 9). A previous study found that peatland WTD deepening benefits shrub dominance while suppresses forbs
and mosses (Mäkiranta et al., 2018). Meanwhile, shrub expansion is reported in Alaska, Siberian and across the pan-Arctic
region under historical climate warming (Tape et al., 2006; Blok et al., 2010). Furthermore, the simulation based on LPJ-
GUESS also predicts higher proportion of shrub NPP in lower latitude regions due to high insolation and deep WTD
(Chaudhary et al., 2020). These studies support our findings that the future warmer and drier condition is the driver for PFT
switch and benefits woody plants.
In PTEM 2.2, the net N mineralization rate is related to soil moisture (Zhao et al., 2022b). Therefore, whether future
peatlands become more nutrient rich depending on the balance between the positive effect of higher temperature and the
negative effect of lower soil moisture. The negative effect of drier soil overwhelms the influence of temperature and thereby
net N mineralization rate decrease under RCP 8.5 after 2100 (SI Figure 8). Under a N limiting condition, the modeling study
with LPX-Bern 1.0 found peatlands switch from a C sink to a source under RCP 8.5 with slow NPP increase, which is
consistent with our simulation with bcc-csm1-1 forcing (Spahni et al., 2013).
Warming affects decomposition mainly in three ways. First, there is higher decomposition rate due to the lower
WTD under warming climate conditions (Huang et al., 2021). Second, higher temperature also enhances decomposition
more than productivity (Tang et al., 2022). Third, in high latitude regions, soil C decomposition rate is likely to increase





under warmer climate and permafrost thaw conditions (Yokohata et al., 2020; Schneider Von Deimling et al., 2015; Gasser
et al., 2018; Macdougall and Knutti, 2016; Schuur et al., 2015). In the warming future, the estimation of $CO_2$ release under
RCP 2.6 tends to be higher than the values estimated from other models (by 2100: 54.7-54.8 Pg C in this study vs. 20-58 Pg
C in literature; by 2300: 131.2-131.3 Pg C in this study vs. 40-98 Pg C in literature, Table 2). However, the estimation under
RCP 8.5 is closer (by 2100: 55.2-57.2 Pg C in this study vs. 42-141 Pg C in literature; by 2300: 222.2-247.6 Pg C in this
study vs. 157-313 Pg C in literature) (Yokohata et al., 2020; Schneider Von Deimling et al., 2015; Gasser et al., 2018) (Table
2). The $CH_4$ emission estimation is also higher than that in Yokohata (2020) by 5-6 Pg C by 2100, while the total C emission
is close to the estimation of MacDougall (2016) (55 Pg C vs. 56 Pg C).

**4.3 Northern peatland C sink and source shift**

Our estimated CAR during 1990-2000 is 19.17-22.73 gC·m$^{-2}$·yr$^{-1}$, which is lower than that by Chaudhary et al.
(2020) during the same period (33.9 gC·m$^{-2}$·yr$^{-1}$). However, our estimated CAR is closer to the core-based Holocene CAR
(18.6-22.9 gC·m$^{-2}$·yr$^{-1}$) (Yu et al., 2009; Loisel et al., 2014). In this study, the estimated pan-Arctic peatlands annual $CH_4$
emissions are 28.7 Tg C·yr$^{-1}$ during 1990-2000, 33.0 Tg C·yr$^{-1}$ during 1990-2000 and 38.5 Tg C·yr$^{-1}$ during 2000-2020. The
estimation after 1990 is close to the 36.0 Tg C·yr$^{-1}$ in Kleinen et al. (2020) while larger than 25.0 Tg C·yr$^{-1}$ reconstructed
from historical data (Treat et al., 2021). However, the difference between our study and Treat et al. (2021) might result from
different peatland coverages used in two studies. Under the peatland coverage of Nichols and Peteet (2019), the CH4
emissions in Treat et al. (2021) were 32.3-43.5 Tg C·yr$^{-1}$, which agrees better with our estimates.
Multiple studies have indicated there is a C loss trend of northern ecosystems under warming climate (Hanson et al.,
2020; Piao et al., 2008; Helbig et al., 2017). In particular, the peatland experiment in Minnesota, USA suggests that warming
increases C loss rate by 31.3 gC·m$^{-2}$·yr$^{-1}$ (Hanson et al., 2020). Similarly, another site-level study on Canadian boreal-
wetland biome shows a decline of $CO_2$ uptake from 25±14 gC·m$^{-2}$·yr$^{-1}$ to 103±38 gC·m$^{-2}$·yr$^{-1}$ by 2100 depending on the
warming scenarios (Helbig et al., 2017). These studies are consistent with our estimates, suggesting that northern peatlands
CAR during 1990-2100 is lower than that during 1940-1990 by 29.1-63.5 gC·m$^{-2}$·yr$^{-1}$.
At the regional scale, whether northern peatlands will switch from a C sink to C source is still uncertain. For
example, Gallego-Sala et al. (2018) indicates northern peatlands are likely to sequester more C under RCP 2.6 and RCP 8.5
until 2100. Chaudhary et al. (2020), however, indicates the C sink capacity of northern peatlands will decreases under RCP
8.5 after 2050. Similarly, McGuire et al. (2018) suggests northern permafrost region could be C sources after 2100 unless
under aggressive climate change mitigation pathways. Furthermore, Qiu et al. (2022) simulates northern peatlands dynamics
until 2300, suggesting a sink-source shift under RCP 8.5 while no such shift under RCP 2.6. Although conclusions vary
among studies, they generally suggest a higher C source possibility under warmer scenarios, which agrees with the negative
correlation between temperature and C sink capacity from this study. Furthermore, the arguments that northern peatlands
keep being C sinks under RCP 2.6 (Gallego-Sala et al., 2018; Qiu et al., 2022) is consistent with our study under bcc-csm1-1
forcing. However, different from previous works (Gallego-Sala et al., 2018; Qiu et al., 2022; McGuire et al., 2018;
Chaudhary et al., 2020), our study predicts northern peatlands to be C sources under RCP 2.6 before 2100 with IPSL-CM5A-





LR forcing. In addition, the C sink-source switch will occur before 2100 under RCP 4.5 and RCP 8.5. Except for the future
decomposition increase, which is common among model predictions (Yokohata et al., 2020; Schneider Von Deimling et al.,
2015; Gasser et al., 2018; Macdougall and Knutti, 2016; Schuur et al., 2015), these differences are mainly due to the
suppressed NPP in this study.
**5. Conclusions**
Northern peatlands responses to future climate change during 1990-2300 are simulated with PTEM. The peatlands
shrink or expansion, peat accumulation and decomposition processes are considered. Two sets of CMIP5 forcing data (IPSL-
CM5A-LR and bcc-csm1-1) are used to drive the model with three warming scenarios (RCP 2.6, RCP 4.5 and RCP 8.5). We
found that wetlands will shrink and permafrost will thaw under all scenarios, indicating pan-Arctic peatlands become
warmer and drier. Northern peatland area expands under RCP 2.6 and RCP 4.5 while shrinks under RCP 8.5 due to high
decomposition rate. NPP does not always increase with temperature because of PFT switch and N limiting effects. However,
both $CO_2$ and $CH_4$ emissions increase with temperature due to lower WTD, thawing permafrost and higher temperature. By
2100, northern peatlands will be a minor C sink of 0.8 Pg C under RCP 2.6 with bcc-csm1-1 forcing while C sources under
other scenarios. During 2100-2300, northern peatlands are C sources under all scenarios, the warmer climate results in the
larger C source. There are negative correlations between temperature and northern peatland C sink under all scenarios. The
negative correlation between precipitation and northern peatland C sink is also found under all scenarios, while this is likely
due to the positive correlation between temperature and precipitation. When pan-Arctic annual temperature is -2.89 - -2.6°C
with IPSL-CM5A-LR forcing or -2.35 - -2.09°C with bcc-csm1-1 forcing, the northern peatlands switch from a C sink to a
source. Similarly, this threshold for annual precipitation is 479.59 - 482.55 mm with IPSL-CM5A-LR forcing and 484.69 -
489.02 mm with bcc-csm1-1 forcing. Our study highlights the current northern peatlands C sink might shift to a source under
future warming and drying climate conditions.
**Acknowledgments**
This study is financially supported by an NSF project (1802832).
**Coda and data availability:**
The data used to reproduce figures in both text and supplementary material, PTEM 2.2 codes, model and samples of running
directory can be accessed via Purdue University Research Repository: https://purr.purdue.edu/publications/4139/1.

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

## Supplemental References

Hugelius, G., Loisel, J., Chadburn, S., Jackson, R. B., Jones, M., MacDonald, G., Marushchak, M., Olefeldt, D., Packalen,
M., Siewert, M. B., Treat, C., Turetsky, M., Voigt, C., and Yu, Z.: Large stocks of peatland carbon and nitrogen are
vulnerable to permafrost thaw, Proceedings of the National Academy of Sciences, 117, 20438,
10.1073/pnas.1916387117, 2020.
Melton, J. R., Chan, E., Millard, K., Fortier, M., Winton, R. S., Martín-López, J. M., Cadillo-Quiroz, H., Kidd, D., and
Verchot, L. V.: A map of global peatland extent created using machine learning (Peat-ML), Geosci. Model Dev.
Discuss., 2022, 1-44, 10.5194/gmd-2021-426, 2022.
Obu, J., Westermann, S., Barboux, C., Bartsch, A., Delaloye, R., Grosse, G., Heim, B., Hugelius, G., Irrgang, A., Kääb, A.
M., Kroisleitner, C., Matthes, H., Nitze, I., Pellet, C., Seifert, F. M., Strozzi, T., Wegmüller, U., Wieczorek, M., and
Wiesmann, A.: ESA Permafrost Climate Change Initiative (Permafrost_cci): Permafrost Climate Research Data
Package v1., Centre for Environmental Data Analysis [dataset],
https://catalogue.ceda.ac.uk/uuid/1f88068e86304b0fbd34456115b6606f, 2020.




Xu, J., Morris, P. J., Liu, J., and Holden, J.: PEATMAP: Refining estimates of global peatland distribution based on a meta-
analysis, CATENA, 160, 134-140, https://doi.org/10.1016/j.catena.2017.09.010, 2018.
Yi, Y. and Kimball, J. S.: ABoVE: Active Layer Thickness from Remote Sensing Permafrost Model, Alaska, 2001-2015,
10.3334/ORNLDAAC/1760, 2020.
Zhao, B., Zhuang, Q., and Frolking, S.: Modeling carbon accumulation and greenhouse gas emissions of northern peatlands
since the Holocene (in review), 2022.
Zomer, R. J. and Trabucco, A.: Version 3 of the "Global Aridity Index and Potential Evapotranspiration (ET0) Database":
Estimation of Penman-Monteith Reference Evapotranspiration. (In Press). [dataset],
https://cgiarcsi.community/2019/01/24/globalaridity-index-and-potential-evapotranspiration-climate-database-v3/, 2022.