# Peer review of "Peatlands and their carbon dynamics in northern high latitudes from 1990 to 2300: A process-based biogeochemistry model analysis"

_EGUsphere, 2022_

## Author Comment (AC1)

**General Comments:**

What is the time-step of the PTEM model?  If it is hourly to daily, I think the water table depth extrapolations are probably meaningful.  However, PTEM is run at an annual time step. The extrapolation of mean water table depth to constant states throughout an annual cycle is inappropriate. The dynamics of winter snowfall accumulation and melt, precipitation variation and temperature, and seasonal changes in evapotranspiration drive a dynamic water table depth in peatlands that needs to be captured in models. With decomposition, methanogenisis and methanotrophy being dependent on oxygenation associate with dynamic water table position.

**Author's response:** The time step of PTEM is monthly. When simulating water table dynamics on monthly scale, we consider the snow melt, precipitation and evapotranspiration which all influence peatland water table position.

Due to the extensive use of abbreviations, I found the manuscript hard to read and follow.  Please limit their use.

**Author's response: W**e deleted the abbreviations for the uncommonly used terms in the paper (e.g., CRU, ALD, FAO), and add Table 1 at the beginning of the Method section for abbreviations and their full names.

While certain science disciplines may be well acquainted with the IPCC forcing  and climate projections, I believe the paper would be more approachable by the average reader if Table S2 and a similar Table for forcing details were included in the published paper.

**Author's response:**  SI Table 1 was added to compare IPSL-CM5A-LR and bcc-csm1-1 forcing.

Arctic regions in the posted polar maps seem to extend into Boreal regions. What land cover is the basis the global calculations of change impact? Perhaps a supporting table defining the lands for the model spatial extrapolations would be helpful.  How much peatland/wetland area? How much upland area? How much permafrost vs. discontinuous permafrost vs. temperate area?

**Author's response: I**n this study, we only consider wetlands/peatlands. When considering the climate change, we are talking about grid cells with peatlands. When considering ecosystem dynamics, we are only talking about the peatlands. There are some peatland coverage in boreal regions, but with relatively low abundance (Figure S2, left panel). The dynamic peatland/wetland area and permafrost area is given in Table S2, and in the main text Section 2.2.3:

'…the pan-Arctic wetlands area for the reference dataset is 2.93 $Mkm^2$, the calibrated wetlands area with IPSL-CM5A-LR forcing input is 2.81 million $km^2$, and with bcc-csm1-1 forcing input is 2.86 million $km^2$.'

The abundance of wetland is presented by Figure S3-4. We do not include information on upland coverage since it's not our research objective. The permafrost extent

is in Figure S6-7. PTEM only simulates one active layer depth for one grid cell. Therefore, PTEM can only classify grid cells into with permafrost or without permafrost. If a grid cell is in the discontinuous permafrost region, i.e., part of the grid cell has permafrost and the other part has no permafrost, it's still classified into these two categories depending on the forcing data. The area of the non-permafrost area is simply the difference between 'northern peatlands permafrost area' and 'total peatland area' in Table S2. We've added rows to Table S2 for this information.

I don't understand the authors' use of the phrase "run-on". Is this suggestive of a transitional fen state between raised bogs and open aquatic systems?

**Author's response:** It refers to the water input from nearby water body or ground water to the peatlands. We've added this explanation to where run-on shows up the first time (Introduction section, line 61).

Does PTEM include depth layers? Are the bins mentioned on Line 254 peat layers?

**Author's response:** PTEM simulates peat thickness for 1cm depth layers. But the bins in line 254 are the bins classified by the TWI values in one grid cell. These bins are essentially dividing the 0.5 degree grid cells into finer grid cells of 0.05 degree resolution (1 into 100). However, these 100 grid cells are derived by clumping a bunch of even smaller grid cells with similar TWI values. We've clarified this by saying '…among the 100 bins classified by TWI…'. To make the technical routine easier to understand, we've added a figure to Supporting Information to explain how to derive water table from 100 TWI bins.

**Specific Suggestions:**

Line 7: Adjust text to: "have been a large C sink…."

**Author's response:** Change has been made.

Line 10: Adjust text to: "Peatland area expansion, shrinkage and C accumulation and decomposition are modeled."

**Author's response:** Text was adjusted.

Line 14: I don't understand "the peatland being C sources".

**Author's response:** Text was adjusted as 'C sources are attributed to'.

Line 24: Adjust text to "plant litter from being fully decomposed"

**Author's response:** Text was adjusted.

Line 31: A possible additional phenology reference:

Richardson AD, Hufkens K, Milliman T, Aubrecht DM, Furze ME, Seyednasrollah B, Krassovski MB, Latimer JM, Nettles WR, Heiderman RR, Warren JM, **Hanson PJ** (2018)

Ecosystem warming extends vegetation activity but heightens cold temperature vulnerability. Nature 560:368-371, doi:10.1038/s41586-018-0399-1.

 **Author's response:** The reference was added.

Line 32:  You might change "will" to would.

 **Author's response:** The word was replaced.

Line 35:  Change to: "To date, multiple modeling studies" to distinguish this work from observational or experimental data.

 **Author's response:** 'modeling' was added.

Line 41:  Change "considering" to including.

 **Author's response:** Word was replaced.

Line 54:  Change "However, the" to ", but future ….".

 **Author's response:** Text was adjusted.

Lines 93 & 100:  What is CRU?

 **Author's response:** CRU is Climate Research Unit, we've removed the abbreviation for CRU and replace it with full name throughout the text.

Line 205:  Please add the specifics of the CO2 concentrations simulated to the treatment tables mentioned above.

 **Author's response:** $CO_2$ concentration was added to Table S2.

Lines 243 to 246:  Some explanation for the inclusion of and need for pH specificity would help the reader.

 **Author's response:** We've added 'pH values are influential to $CH_4$ production process…' to explain why we simulate pH values.

Line 290:  Expand "the threshold temperature" to "the threshold temperature needed to transition the peatland from a C sink to a C source".  If you state it this way, lines 291 and 292. Might not be needed.

 **Author's response:** We've changed text to 'the threshold temperature needed to transition the peatland from a C sink to a C source'. However, we think it's better to keep line 291-292 because they explain how the thresholds are calculated.

Page 12:  I would like to see the quantitative specifics provided in Supplement Tables S3 and S4 brought into the main manuscript.

Author's response: These two tables were moved to the main text.

Page 14:  A very nice paragraph to describe conditions before 2100.

Table 1.  It is a bit odd to suggest that a transition from one frozen state to another can have large impacts. This interpretive oddity results from the use of mean annual values.  Is there another variable that could also be included?  For example, days above freezing that could perhaps better explain the results.

Author's response: We've added threshold unfrozen day number to Table 1 (now Table 4). We also explained the method we use to estimate daily temperature is derived from a previous work in Section 2.3.3. In addition, a SI figure 22 was added to show the time series of unfrozen day number.

Line 414:  Change "no" to not.

Author's response: 'no' was replaced by 'not'.

Section 4.2 Recent experimental results showing the relationship between warming and nutrient availability could be cited here:

Iversen CM, Latimer J, Brice DJ, Childs J, Vander Stel HM, Defrenne CE, Graham J, Griffiths NA, Malhotra A, Norby RJ, Oleheiser KC, Phillips JR, Salmon VG, Sebestyen SD, Yang X, Hanson PJ (2022) Whole-ecosystem warming increases plant-available nitrogen and phosphorus in an ombrotrophic bog. Ecosystems https://doi.org/10.1007/s10021-022-00744-x

Author's response: We've discussed and cited this article in Section 4.2.

Line 463:  The units for Hanson et al. 2020. Data are wrong they should be gC $m^{-2}$ $y^{-1}$ $°C^{-1}$.  The implication is that the Hanson et al. results are proportionate to the amount of warming.

Author's response: We've changed the text into 'each 1°C of warming increases C loss rate by 31.3 $gC \cdot m^{-2} \cdot yr^{-1}$.

---

## Author Comment (AC2)

General Comments:

I feel that a potential reader will benefit from inclusion of a study area (region) map to give context to the scale and geographical context of the region of interest. I particularly enjoyed seeing the maps in the Supplementary Information file and a map depicting this area will greatly benefit this manuscript.

**Author's response:** We've moved SI Figure2 to the main text. SI Figure2 shows the extent of wetlands in 1940-1990, which is the initial wetland extent for future simulation.

I find that the text relating to your workflow methodology is confusing, this will be greatly improved if the authors include a simplified figure of the steps taken in the analysis. In the current format, the workflow methodology is difficult to follow and potentially replicate and test in future studies.

**Author's response:** We've divided the workflow into two parts and simplified each part. To help understanding, we also added a SI Figure 1 to explain the way we estimate wetland extent dynamics.

Some parts of the manuscript lack clarity and careful attention needs to be given to this.

The use of multiple abbreviations makes the manuscript difficult to read in places where multiple abbreviations are used concurrently.

**Author's response:** We deleted the abbreviations for the uncommonly used terms in the paper (e.g., CRU, ALD, FAO), and added Table1 at the beginning of the Method section for abbreviations and their full names.

Please provide context to the reader when referring to "run-on".

**Author's response:** It refers to the water input from nearby water body or ground water to the peatlands. We've added this explanation to where run-on shows up the first time (Introduction, line 61).

Specific Comments:

Line 7: This sentence lacks clarity, have they been a C sink during the entire Holocene?

**Author's response:** The text was changed to 'have been a large C sink'.

Line 14: Adjust the text to "C sources are attributed to 1) peatland water table depth..."

**Author's response:** The text was changed as suggested.

Line 19: It will be beneficial to the reader to indicate specifically how much ealier this shift will take place.

**Author's response:** We've specified 'This study predicts an northern peatland sink to source shift in around 2050, earlier than previous estimates of after 2100 in the literature…'.

Line 22: Please provide an in-text reference for the primary reference of this definition of peatland ecosystems.

**Author's response:** The reference was added.

Line 29: Please provide details on the phenology of peatlands in this region.

**Author's response:** We've added a reference on phenology and specified 'triggering spring onset earlier and delaying autumn green-down'.

Line 29: The authors use the word "severe", rather quantify (increase in °C) being currently measured or projected between the Arctic region and surroundings. This needs more detail.

**Author's response:** We've added details by saying 'the warming in the Arctic region is almost three times as much as the global average'.

Lines 28-35: It is unclear whether these changes have been observed already (and have been quantified) or they are projected climate warming influences. As the previous comment suggests, please be more specific here. This detail will add to your argument.

**Author's response:** We've added details (bolded words):

Peatlands are vulnerable to disturbances induced by climate warming (Loisel et al., 2021), especially when the warming in the Arctic region **is almost three times as much as the global average** ( GISTEMP-Team., 2021). First, warming influences northern terrestrial ecosystem vegetation productivity by increasing spring photosynthesis, **triggering spring onset earlier and delaying autumn green-down** (Piao et al., 2008; Helbig et al., 2017, Richardson 2018). Second, warming could induce drier Arctic conditions **with 21% of lake count and 2% of lake area decrease found during 1960s-present** (Finger Higgens et al., 2019), and peatlands water table drawdown would result in **net increase of greenhouse gas emissions of 0.86 Gt $CO_2$-eq $\cdot yr^{-1}$ by 2100** (Huang et al., 2021). Third, decomposition rate increases under higher temperature and previous studies found positive linear correlations between warming and **net C loss rate of 31.3 $gC \cdot m^{-2} \cdot year^{-1} \cdot °C^{-1}$** (Hanson et al., 2020). Fourth, permafrost thaw under warming conditions will expose previously-frozen C for **dissolving and decomposition** (Gandois et al., 2019)

Line 88: What about the peatland dynamics? This sentence must be rephrased as it is misleading.

**Author's response:** We've rephrased the sentence to 'Although this study aims at the peatland C expansion, shrinkage, accumulation and decomposition after 1990…'

Figure 1: Within the figure boxes please include the reference (citation) of the different datasets used. This will also aid in describing your workflow.

**Author's response:** We've added the citation to the workflow.

Line 414: Adjust the text from "no" to not

**Author's response:** 'no' was replaced by 'not'.

Line 436: Adjust the text from "depending" to depends.

**Author's response:** 'depending' was replaced by 'depends'.

---

## Author Comment (AC3)

General Comment:

If I understand it correctly, the authors first use TOPMODEL to simulate wetland extent, then wetland areas with peat thickness>=30 cm are defined as peatland. However, the distinction of wetlands and peatlands in the paper are fuzzy. For example, in SI Figure2, it compares wetland abundance with peatland maps, an evaluation of simulated wetland area is missing. The paper sometimes confuses wetlands and peatlands in the main text.

**Author's response:** We've added the explanation to Section 2.2.3:

"Notably, the extent of pan-Arctic peatlands is used as an approximation of pan-Arctic wetlands because the northern peatland extent is estimated to be 2.9-3.3 Mkm$^2$, with an average of 3.05 Mkm$^2$ (Xu et al., 2018; Hugelius et al., 2020; Melton et al., 2022); while the northern wetland extent is estimated to be 3.2 Mkm$^2$ (Olefeldt et al. 2021), indicating northern wetlands are dominated by northern peatlands. In addition, the peatland coverage from Xu et al. (2018) and Hugelius et al.(2020) both include the shallow peats (<30cm), which is classified as wetlands rather than peatlands in this study."

Is peatland resilience considered in the model? Lowering of water table and enhanced decomposition can lead to compaction of surface peat, with a lower hydraulic conductivity, it helps maintain soil moisture and slow down decomposition.

**Author's response:** This effect is not considered in PTEM.

It is hard to read the manuscript due to too many abbreviations, and too many procedures to make the start of new simulations consistent with the Holocene simulation.

**Author's response:** We've divided the workflow into two parts and simplified each part. To help understanding, we also added a SI Figure 1 to explain the way we estimate wetland extent dynamics. In addition, we deleted the abbreviations for the uncommonly used terms in the paper (e.g., CRU, ALD, FAO), and added Table1 at the beginning of the method section for abbreviations and their full names.

Specific Comments:

L53-54: peatland extent in Qiu et al., 2020, Müller and Joos, 2021 are not fixed. Both studies dynamically simulated peatland area in the future.

**Author's response:** We've deleted these two citations here.

L62-63: Incomplete sentence, when peat thickness is ?

**Author's response:** We've rephrased this sentence to 'and the theoretically maximum run-on corresponding with 0 cm peat thickness'.

Eq.9: Why moss has been excluded from the calculation of evapotranspiration? Is it because the FAO algorithm and Penman-Monteith model are not applicable to moss? Anyway, moss shouldn't be neglected not only because of their high abundance in peatland, but also because of the special characteristics of moss: due to the lack of stomata, moss cannot control the water loss to the atmosphere.

**Author's response:** For one reason, we haven't seen works applying FAO algorithm to moss. For the other reason, evapotranspiration in PTEM is calculated as how much water is extracted from the soil, so the non-vascular plants does not fit into this category. Since the top hydrology layer in PTEM is a moss layer, the evapotranspiration from moss layer is better represented by the soil evaporation from the top hydrology layer. We find a mistake in the original text saying E_soil is the evaporation from bare soil – actually, E_soil is the evaporation from the top hydrology layer (moss). We've corrected this.

L254-255: If the 50-year running mean WTD for a bin is shallower than 25cm for only one time during the whole 1990-2300 timeseries, then this bin is "potential peatlands"? Would be informative if you could add a figure to show the frequency distribution of long-term WTD of all bins/grid cells.

**Author's response:** That's correct. Two SI figures (6&7) were added to the appendix on the average water table depth of the simulated bins. These two figures are cited in result Section 3.1.

L302: higher than what? Should be "increasing temperature and precipitation"?

**Author's response:** We've changed 'higher' to 'increasing'.

L310: stable throughout 1990-2300? the increase in temperature obviously slows down after 2200

**Author's response:** We've rephrased the sentence to '…is stable during 1990-2300, and slows down after 2200'

SI Figure9: Even at present-day, 1990-2000, pan-Arctic peatlands are dominated by woody PFTs? Is it realistic? A validation of simulated vegetation composition is missing in this manuscript.

**Author's response:** Yes, the fraction of woody plants could be overestimated. One explanation is the 'potential peatlands are included, where the WTD is usually lower than the existing peatlands and is more suitable for woody plants'. We've added this to Discussion section 4.2, and cited a previous PTEM simulation when PFT is only considered for the existing peatlands. In this previous simulation (Zhao el al. 2022 (b)), the fraction of herbaceous plants is much higher than presented in this study (SI Figure 10 & 16 in Zhao el al. 2022 (b)).

L358: In Figure4 3(a), CO2 emission (net ecosystem exchange, NEE?) are positive over 1990-2300 for both bcc-csm1-1 and IPSL-CM5A-LR, that means pan-Arctic peatlands are C sources during 1990-2100 under bcc-csm1-1 RCP 2.6, isn't it?

**Author's response:** The $CO_2$ emissions are the amount of $CO_2$ released by soil heterotrophic respiration. We've clarified this in the caption by adding '$CO_2$ emissions from soil heterotrophic respiration'. Whether peatlands are C sinks or sources are determined by the trend of the top panel (C stocks).

L446-447: I couldn't find Table 2 in the manuscript.

**Author's response:** Table2 was added (now Table 5).

L487-488: Isn't WTD a more direct driver for peatland area change? Why high decomposition rate result in peatland area shrinks?

**Author's response:** Peatland here is defined as area with peat over 30cm. Under high decomposition, the peat thickness decreases, and peat area over 30cm also decreases. We've clarified this process by saying '…due to the shrinkage of the area with over 30cm peat thickness under high decomposition rate'.